# Phylogenetic and Molecular Analysis of the Porcine Epidemic Diarrhea Virus in Mexico during the First Reported Outbreaks (2013–2017)

**DOI:** 10.3390/v16020309

**Published:** 2024-02-18

**Authors:** José Francisco Rivera-Benítez, Rebeca Martínez-Bautista, Raúl González-Martínez, Jazmín De la Luz-Armendáriz, Irma Herrera-Camacho, Nora Rosas-Murrieta, Laura Márquez-Valdelamar, Rocio Lara

**Affiliations:** 1Centro Nacional de Investigación Disciplinaria en Salud Animal e Inocuidad, Instituto Nacional de Investigaciones Forestales, Agrícolas y Pecuarias, Mexico City 04010, Mexico; 2Zoetis Swine, Mexico City 05120, Mexico; rebeca.martinez@zoetis.com; 3Cargill, Mexico City 01210, Mexico; raul_gonzalez@cargill.com; 4Departamento de Medicina y Zootecnia de Rumiantes, Facultad de Medicina Veterinaria y Zootecnia, Universidad Nacional Autónoma de México, Mexico City 04510, Mexico; delaluzarmendarizj@fmvz.unam.mx; 5Laboratorio de Bioquímica y Biología Molecular, Centro de Química, Instituto de Ciencias, Benemérita Universidad Autónoma de Puebla, Puebla 72000, Mexico; irma.herrera@correo.buap.mx (I.H.-C.); nora.rosas@correo.buap.mx (N.R.-M.); 6Laboratorio de Secuenciación Genómica de la Biodiversidad y de la Salud, UNAM, Mexico City 04510, Mexico; lmarquez@ib.unam.mx; 7Programa de Maestría en Ciencias de la Producción y de la Salud Animal, Facultad de Medicina Veterinaria y Zootecnia, Universidad Nacional Autónoma de México, Mexico City 04510, Mexico

**Keywords:** porcine epidemic diarrhea virus, phylogenetic analysis, spike protein

## Abstract

The characteristics of the whole PEDV genome that has circulated in Mexico from the first outbreak to the present are unknown. We chose samples obtained from 2013 to 2017 and sequenced them, which enabled us to identify the genetic variation and phylogeny in the virus during the first four years that it circulated in Mexico. A 99% identity was found among the analyzed pandemic strains; however, the 1% difference affected the structure of the S glycoprotein, which is essential for the binding of the virus to the cellular receptor. The S protein induces the most efficacious antibodies; hence, these changes in structure could be implicated in the clinical antecedents of the outbreaks. Antigenic changes could also help PEDV avoid neutralization, even in the presence of previous immunity. The characterization of the complete genome enabled the identification of three circulating strains that have a deletion in ORF1a, which is present in attenuated Asian vaccine strains. The phylogenetic analysis of the complete genome indicates that the first PEDV outbreaks in Mexico were caused by INDEL strains and pandemic strains related to USA strains; however, the possibility of the entry of European strains exists, which may have caused the 2015 and 2016 outbreaks.

## 1. Introduction

Porcine epidemic diarrhea (PED) was first described in England in 1971. In 1978, the prototype strain CV777 was characterized in Belgium [1]. The etiological agent of PED is an RNA single-stranded virus, positive-sense, of the Alphacoronavirus genus. The genome’s length is approximately 28 kb, and it contains two untranslated regions (UTRs) located at the 5′ and 3′ terminals, seven open reading frames (ORFs) (ORF1a-ORF1b-S-ORF3-E-M-N) that code for structural proteins (S, E, M, and N), and three non-structural proteins (ORF1a, ORF1b, and ORF3) [2,3]. The products coded by ORF1a and ORF1b produce 16 non-structural proteins (NSPs), which constitute the replication and transcription complex. NSP3 is located within the ORF1a polyprotein and corresponds to the largest protein of the coronavirus genome, possessing diverse multidomains with functions that are related to the suppression of the immune response of the host and replication and transcription processes [4]. In this region, a large variability in PEDV strains is found [5], and it has been determined that mutations and deletions are produced in this protein during the in vitro attenuation process [6,7]. The spike protein (S) is a type I glycoprotein divided into two domains. The S1 domain, in which the N-terminal domain (NTD) is found, corresponds to the globular part of the protein related to binding to the cellular receptor (porcine amino peptidase N) [8]. Thus, it is the most variable region of the genome, and its relevance lies in that it presents diverse neutralizing epitopes known as COE (equivalent to the collagenase of the transmissible gastroenteritis virus) [3,9,10]. The S1 domain has insertions and deletions that have given rise to the strains known as S-INDEL, which are characterized by lower virulence compared to global pandemic strains [11]. The S2 domain corresponds to the trunk of the glycoprotein. It is related to the fusion of the virus to the host’s cell and it is conserved in comparison to the S1 domain; thus, by having diverse immunodominant neutralizing epitopes, it could be used as an immunogen [3,12]. The structure of the S glycoprotein has been studied through the mouse hepatitis virus (MHV) model to analyze its organization, and the S1 region corresponds to a multidomain. The S1-NTD region contains domain A, located in the first 300 amino acids (aa), and domains B, C, and D are located in the S1-CTD region. Domain C corresponds to the binding site of the receptor in the case of the PEDV. Structural variations confer the molecular bases to determine the specificity of coronaviruses and their tropism [13]. The ORF3 protein is related to the virulence of PEDV [14], because during the in vitro attenuation process, an early end is produced that truncates the protein [15], thereby decreasing ionic channel activity [16]. Ionic channels are important because they are related to the regulation of replication, entry, assembly, and release of the virus [17]. Protein E is responsible for the assembly of the virion, and in co-expression with protein M, the proteins form pseudoparticles that are capable of interfering with the immune response of the host [2,3,14]. Protein M, aside from intervening in the virus formation process, induces neutralizing antibodies in the host and interferes with the production of interferon-like protein N, which is a phosphoprotein that associates with the viral genome [2,3,14]. PEDV induces severe watery diarrhea and vomiting in pigs of all ages; however, mortality can reach 100% in piglets [18]. These same clinical signs occur in outbreaks caused by another Alphacoronavirus, transmissible gastroenteritis virus (TGEV), of pigs [19]. However, genetically, PEDV has a higher identity and phylogenetic relationship to the coronavirus of the bat, BtCoV/512/2005 [20]. Molecular biology and phylogenetic analyses have been based mainly on proteins S and M, ORF3, and the complete genome. By means of the S protein, it has been possible to classify PEDV into three main genogroups: G1, including the classical and Asian strains, G2, including strains circulating globally since 2010, and INDEL [14,21]. In Mexico, molecular studies have determined the circulation of strains G2b and INDEL with homology to the North American strains, and the in silico analysis of the S protein showed that there are significant changes in binding sites to the receptor that could be related to recurring outbreaks [21]. In studies carried out in Mexico by other research groups, the circulation of pandemic PEDV strains has been identified and in no case have INDEL strains been identified [22,23,24]. The study of the complete genome of the North American strains has classified PEDV in classical, pandemic, and INDEL strains [5,25]. However, insufficient information is available to perform molecular epidemiology and phylogenetic studies of PEDV in the reported early outbreaks. The objective of the present study was to sequence the complete genome of PEDV from the first detected outbreaks of the disease in 2013 until the recurring outbreaks of 2017, to understand the origin of currently circulating strains, comparing them with global strains, and to characterize the genome.

## 2. Materials and Methods

### 2.1. Sample Collection

We obtained 211 samples from the intestine and feces of piglets with enteric clinical disease suggestive of PED from 31 swine farms located in seven states of the Mexican Republic (Jalisco, Puebla, Veracruz, Sonora, Guanajuato, Michoacán, and Querétaro). Veterinarians and pig farmers voluntarily provided the samples used in this study. The study was conducted according to the guidelines of and requirements indicated by the Institutional Bioethics Committee for the Care and Reasonable Use of Experimental Animals in Research Projects CENID-Microbiología Animal (Approval Code: CBCURAE-007-12/15/2015). Samples were collected from July 2013 to June 2017 and were kept at −70 °C until processing. 

### 2.2. RNA Extraction and cDNA Synthesis

RNA was extracted using the RNeasy MiniKit (QIAGEN) following the manufacturer’s protocol. Real-time qRT-PCR was used to determine to which genogroup the samples belonged, INDEL or NO INDEL [26]. Next, we chose the samples based on the highest number of genomic equivalents per state and year of sampling. Once the samples had been selected, the synthesis of cDNA was performed using the RevertAid First Strand cDNA Synthesis kit (Thermo Fisher Scientific, Waltham, MA, USA).

### 2.3. Amplification of the PEDV Genome and Sequencing

To amplify the PEDV genome, we designed 12 pairs of primers using sequence MEX/124/2014 (GB KJ645700) (Table 1) as the base. With these primers, the PCR reactions were standardized for the amplification of the complete genome using the Phusion Green Hot Start II High Fidelity DNA Polymerase kit (Thermo Fisher Scientific). Amplification conditions were 98 °C for 30 s, 35 cycles of 98 °C for 10 s, 60 °C for 30 s, 72 °C for 90 s, and a final extension of 72 °C for 10 min. Sequencing was performed using the platform Ion Torrent PGM, and the ensemble of genomic sequences were generated using the Galaxy tools (https://usegalaxy.org/).

### 2.4. Analysis of Sequences and Phylogeny

Using the complete genome of the 24 strains obtained in the present study and 59 contemporary and ancestral PEDV strains available in GenBank, which were reported in Asia, the Americas, and Europe, we performed an alignment of nucleotides and of sequences of amino acids through ClustalW using the software MEGA 7 (v.7.0.21). Based on the alignments and by means of MEGA 7, we determined the best model for the construction of the phylogenetic analysis using the maximum likelihood method, and the feasibility of the internal branches was evaluated by 1000 bootstrap repeats. Additionally, an analysis of 24 PEDV Spike sequences reported in GenBank only in Mexico, between the years 2013 and 2018, has been included (there are no sequences reported after this year). The phylogenetic analysis were carried out as previously described.

### 2.5. Trimeric Spike Protein Structure Prediction

The amino acid sequences of protein S were analyzed to determine structural differences among the PEDV strains. For this, molecular modeling was performed by means of the Swiss-Model server (Swiss Institute of Biotechnology), using the homotrimeric spike protein of the human coronavirus NL63 as a template [27]. Prediction of the structure of protein S was visualized and analyzed by means of the PyMOL program.

## 3. Results

### 3.1. Analysis of Sequences

Of the samples collected from 2013 to 2017 and analyzed with real time RT-PCR, reported by Wang et al. [26], we determined that 78.2% of samples were positive (25 positive samples corresponding to the INDEL genotype and 140 to pandemic); in two farms, a co-infection with pandemic and INDEL strains was detected. The pandemic strains had a higher number of genomic equivalents in all co-infections. Of all of the positive samples, 24 were selected for amplification and sequencing (Table 2), and they had an average depth of 713×. By means of alignment, it was determined that strain PEDV/MEX/GTO/02/2016 has a deletion of 24 nucleotides in the ORF1a region, which represents a deletion of eight amino acids in the protein. In the strains detected in Querétaro in 2017, a deletion of six amino acids was recorded in the region (Figure 1). In the S gene, two strains (PEDV/MEX/03/JAL/2016 and PEDV/MEX/JAL/19/2017) had an insertion of three amino acids (232LGL234). The identity among the pandemic strains circulating from 2013 to 2017 in Mexico was higher than 99.1%, whereas with the INDEL strains, the homology was 98.9%.

### 3.2. Phylogenetic Analysis

Phylogenetic analyses were performed based on the alignment of 83 PEDV sequences. A tree was constructed from the complete genome, which allowed the identification of a clade of pandemic strains. The pandemic strains were divided into two clades: clade one, including 20 Mexican clades collected between 2013 and 2017, and clade two, including the PEDV/MEX/SON/01/2015 strain. Three strains from this study are within the INDEL clade, related to two subclades. This analysis showed that the Poltatova01/2014 strain from Ukraine gave rise to a subclade that included five Mexican strains that caused PEDV outbreaks between 2015 and 2017, indicating that there was probably an entry of European PEDV strains into Mexico or that genetic mutations were the same in these regions (Figure 2A). The phylogeny of the S glycoprotein corroborates that the first outbreaks in Mexico were related to USA-origin strains and that the INDEL strains have been circulating in Mexico since 2013 (Figure 2B). Due to the deletions in the ORF1a region in three Mexican strains, we studied the phylogeny of polyprotein 1a, and we observed that these strains are located in a subclade together with another Mexican strain that does not have deletions in this region. However, the strains are genetically distant from the attenuated vaccinal Asian strains. This same analysis allowed the identification of the pandemic clades. Strain PEDV/MEX/SON/01/2015 is located in subclade two, together with a pair of Mexican strains reported in 2014 (KJ645700 and KR265766) and a Canadian strain from Quebec (KR265831), according to the analysis of the complete genome (Figure 2C). The analysis of ORF3 classified PEDV into two main clades, which grouped the attenuated Asian PEDV strains into one clade corresponding to strains with a truncated ORF3 protein (Figure 2D).

For the analysis of the Spike sequences reported exclusively in Mexico, three genogroups were identified. The genogroups correspond to G2a, G2b, and INDEL. The 24 sequences reported in the GenBank by different research groups are grouped in G2b, and in no case had strains from G2a or INDEL been identified in the years 2013 to 2018. The phylogenetic tree is presented in Appendix A.

### 3.3. Prediction of the Structure of the Homotrimeric Spike Protein

We were able to predict the secondary structure of the trimeric form of the proteins of strains from Colorado/2013 (prototype), PEDV/MEX/MICH/02/2015 (INDEL), and PEDV/MEX/JAL/19/2017 (pandemic) using the glycoprotein of the human coronavirus NL63 as a template with an identity of 45%. It was evident that the proteins of the pandemic and INDEL strains have a structural difference in domain A (shown in cyan). In the pandemic strains, there is a region of the amino acids sequence (59QGVNST64) that forms the alpha helix, whereas in the INDEL strains, turns are formed (55SMNSSS60), which are located in a region of deletions. This same region is conserved among the Mexican strains analyzed in this study, except in strain PEDV/MEX/JAL/19/2017, where substitution N58K yields a change in the alpha helix, making it longer. In this same domain A, additional differences were observed, in which S proteins of the INDEL strains generate an alpha helix and a beta sheet in the 126DNKTL130 and 223YYE225 regions, respectively, whereas in the pandemic strains, there is a turn and an alpha helix (130SIKTL134 and 226SYQ228). Insertion 232LGL234 (marked in red) is located in the linker region, placed between domain A and B, modifying the structure of the protein. The structural analysis of the COE region (shown in blue) revealed that only substitutions of the protein in strain PEDV/MEX/JAL/19/2017 change the structure to a beta sheet (561YGYVSK566), where originally the prototype strains of Colorado/2013 had an alpha helix (564FGYVSN569). Changes in region S2 of the S glycoprotein do not affect structural conformation (Figure 3A). Using PyMOL software, the electrostatic potential of the trimer of the S glycoprotein was determined to identify the changes produced by the substitutions of amino acids based on the sequence of Colorado/2013. Differences were found in the domain A region of the protein in strain INDEL PEDV/MEX/MICH/02/2015, such as the D175H substitution, where a change in a negative residue to one with a positive charge occurred. Another example is the E206T substitution, neutralizing the zone where this change occurs. For strain PEDV/MEX/JAL/19/2017, the N58K change, aside from affecting the alpha helix structure, turns the charge positive, whereas the 232LGL234 insertion changes the structure and maintains the region without a charge. In domain B, a decreased number of changes was identified, as is the case of N351D in the INDEL strains and H369R in the 2017 pandemic strain. However, diverse changes in polarity and charge exist in the COE region, and only the K569N substitution is seen in strain PEDV/MEX/JAL/01/2017 (Figure 3B). Changes in the sequence of domain A and the COE region are shown in Figure 3C,D.

## 4. Discussion

Until 2017, swine producers in Mexico reported PED outbreaks in which the mortality of piglets less than 1 week of age reached 100%, despite immunity to PEDV and control strategies (including vaccination and feedback). Economic losses are related to the high mortality in piglets and to the diarrhea caused in weaning, growing and fattening stages. For example, Schulz & Tonsor [28] described a 43% decrease in the number of pigs sold per sow in 2014 in the USA. It has also been reported that economic losses are related to the decrease in the reproductive parameters of sows, such as a decreased number of live-born piglets, a lower weight at birth, and increases in the rate of return to estrus and in the abortion rate [29]. The region of the ORF1a polyprotein, where deletions were found in three Mexican strains, corresponds to the domain “papain-like protease 1 (PLP1)” of the protein NSP3, which contains diverse domains implicated in the replication–transcription complex of the genome and implicated as antagonists of interferon induction [4]. However, in contrast to the attenuated Asian strains, where no severe diarrhea is produced, the Mexican strains resulted in high rates of piglet mortality. The PLP1 domain, analyzed using the model of transmissible gastroenteritis virus (TGEV), was found to be involved in the innate immune response and proinflammatory signaling [30]. Attenuation of a strain cannot be evaluated with only one region of the genome, as mentioned by Chen et al. [15], who performed PEDV adaptation in cell cultures and analyzed the genome of the attenuated strains. Despite the four-year difference among some of the strains collected in this study, an identity of 99% was determined, which indicates that the mutation rate of PEDV is low, as has been reported in studies where the estimated evolution rate is 6.2 × 10^4^ substitutions/site/year for the complete genome. The mutation rate is 1.5 × 10^3^ substitutions/site/year in the most variable portion of the genome, corresponding to domain S1 [5]. The phylogenetic analysis of the complete genome allowed the classification of 20 strains in the present study into two pandemic subclades together with strains that have caused acute outbreaks and high mortality in other countries in the Americas, Asia, and Europe in the same years. These same clades had been identified previously in a study performed by Vlasova et al. [25], where two North American clades were identified. In studies reported in 2015 and 2016, Asian sequences of PEDV that were closely related to the American strains were identified [5,31]. In Mexico, circulation of the first INDEL strain, isolated in 2014, was confirmed [21]. However, in the present study it was determined that the INDEL strains in Mexico had been circulating since 2013 and were grouped in a subclade separate from the USA strains. The phylogeny of the ORF1a polyprotein did not show a direct relationship between the Asian vaccine strains, which have a deletion in region NSP3, and the Mexican strains. This finding may be observed because the Asian attenuated strains originated from classical strains (G1). By means of the prediction of the structure of the trimeric S glycoprotein, it was possible to identify that insertion 232LGL234 in strains PEDV/MEX/JAL/03/2016 and PEDV/MEX/JAL/19/2017 is located on the surface of the protein and that it changes the structure, which can be related to the increase in virulence and the high mortality in the farms affected by these strains, despite existing previous immunity to PEDV. This same situation occurred in South Korea, where strain KNU-1601 was reported to have five insertions in the S1 region of PEDV, which caused severe diarrhea and high mortality in a farm where PEDV immunity had been acquired with vaccination and feedback [32]. Despite these insertions, these strains continue to be grouped in pandemic clades. These same Mexican strains with insertions have a significant change in structure in the binding region to the receptor, and therefore in recognition by neutralizing antibodies. In 2017, it was reported that a similar strain (PEDV/MEX/JAL/01/2016) had a potential antigenic site that was not found in the other circulating strains [21]. In a study carried out in Mexico, 68 samples from 17 states were analyzed, with 77% of the samples being identified as positive [24]. After sequencing, substitutions were evident in the COE region of the Spike protein, of which in two out of three cases they were different from those found in the present study. Domain A of coronaviruses binds to sialic acid receptors [33], and it has been proven that PEDV recognizes Neu5Ac as a co-receptor [34]. The differences between the INDEL and pandemic strains in domain A could be involved in the difference in virulence and PED presentation, since it has been shown that INDEL strains produce less severe diarrhea than pandemic strains [11]. By means of molecular modeling, it was demonstrated that domain A of the S glycoprotein presents structural and charge differences between pandemic and INDEL strains, even in strain PEDV/MEX/JAL/19/2017. The study of this region is relevant to determine the virulence of a strain, as it has been reported that just one change in amino acids in this region or in the binding region to the receptor can significantly alter the virulence of a strain [35]. It has been demonstrated that monoclonal antibodies produced in INDEL domain A were neutralizing against homologous strains, but were not completely neutralizing to pandemic strains [10]. From the previously mentioned results, it can be inferred that pandemic strains have evolved to achieve a more efficient interaction with the receptors of the host cell, causing more severe clinical signs compared to the classical and INDEL strains.

## 5. Conclusions

In conclusion, we molecularly characterized the PEDV genome and its phylogeny, indicating that, despite the high identity between the 2013 and 2017 pandemic strains, it is possible that the changes affect the clinical presentation of the disease and that recurring DEP outbreaks might occur even in the presence of previous immunity. The identification of three strains with deletions in the ORF1a region indicates the viability of future in vitro attenuation studies to attain an attenuated strain that can be used as a vaccine capable of producing humoral and cellular immunity. Based on the findings, molecular characterization of PEDV must be used for implementing disease control strategies.

## Figures and Tables

**Figure 1 viruses-16-00309-f001:**
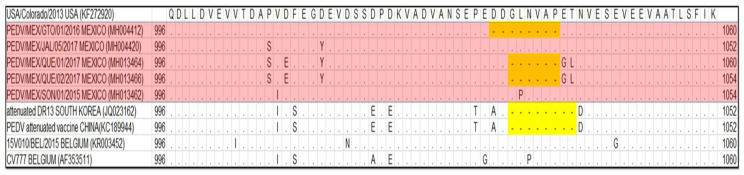
Alignment of the ORF1a polyprotein corresponding to a fragment of the papain-like protease 1 and the location of insertions in the Mexican and Asian attenuated strains. The Mexican strains are highlighted in red.

**Figure 2 viruses-16-00309-f002:**
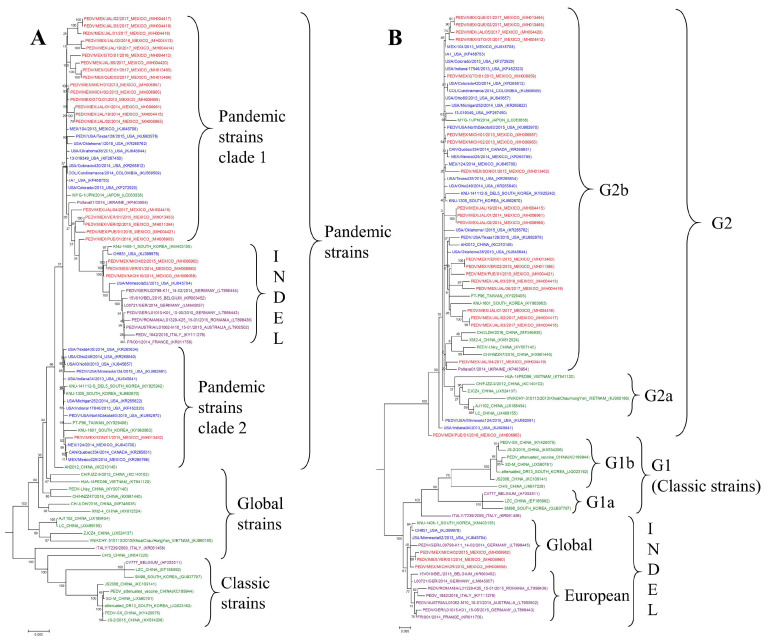
Phylogenetic analyses of PEDV constructed by means of a maximum likelihood algorithm and 1000 bootstraps. The strains from this study are in red, the USA strains are in blue, the European strains are in purple, and the Asian strains are in green. (**A**) Phylogenetic tree of the PEDV genome based on the model GTR + G + I (scale bar: 0.005). (**B**) Phylogenetic tree of the S protein constructed with the model JTT + G (scale bar: 0.005). (**C**) Phylogenetic tree of the ORF1a polyprotein constructed with the model JTT + G + F (scale bar: 0.01). (**D**) Phylogeny of the ORF3 protein constructed using a WAG model (scale bar: 0.05).

**Figure 3 viruses-16-00309-f003:**
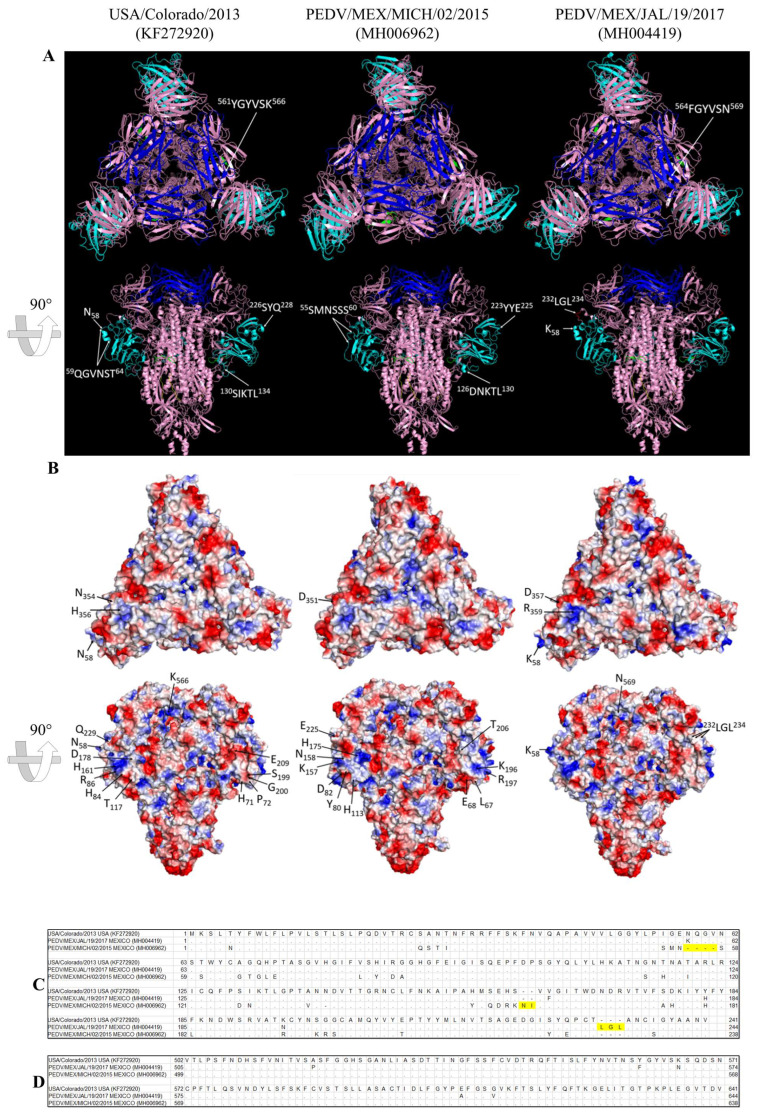
Model of the PEDV S protein trimer and amino acid alignment. (**A**) Model of the secondary structure of the S glycoprotein. Blue corresponds to the COE region and cyan to domain A. In strain PEDV/MEX/JAL/19/2017, the insertion ^232^LGL^234^ is shown in red, the epitope SS” is shown in green, and SS6 is shown in yellow. (**B**) The trimeric structure of the S glycoprotein surface, where the electrostatic potential and the significant changes among the prototype strain (USA/Colorado/2013), the Mexican INDEL strain (PEDV/MEX/MICH/02/2015), and a 2017 Mexican pandemic strain (PEDV/MEX/JAL/19/2017) are shown. (**C**) Alignment of amino acids of the domain A region. (**D**) Alignment of amino acids of the COE region. Insertions and deletion are marked in yellow.

**Table 1 viruses-16-00309-t001:** Primers designed to amplify the complete genome of PEDV.

Primer	5′-3′ Sequence	Amplified Region of the PEDV *	Size bp
PEDv-WG-1F	TCTAGTTCCTGGTTGGCGTTC	136–2527	2392
PEDv-WG-1R	CGTGCCACAGTGACAACAATG		
PEDv-WG-2F	TGGTAGCATCTGGCGGTCTT	2420–4912	2493
PEDv-WG-2R	CCAGCAGGCACTGTTTTGTTA		
PEDv-WG-3F	CTTTCAGGATTGAAGGTGCTCA	4711–7327	2616
PEDv-WG-3R	GACAAACTGGCCAACAACGC		
PEDv-WG-4F	GGTCCAGGCTGCACTTTTAT	6975–9575	2600
PEDv-WG-4R	TTCACCCGGTCTAACTGTGC		
PEDv-WG-5F	TACTGTTATCTGCCCACGCC	9350–11815	2466
PEDv-WG-5R	ATACGCTCACACCCAAGGAC		
PEDv-WG-6F	ACTTGGCAAAGGATGGGGTT	11566–14154	2588
PEDv-WG-6R	GTGGGCAGGATGTTACGCTT		
PEDv-WG-7F	AGCTCGCGTCGTGTATCAAA	13945–16336	2392
PEDv-WG-7R	GCGTGAACATTTGTCATTGCTA		
PEDv-WG-8F	GCAGCGGTCGATTCACTTTG	16274–18639	2365
PEDv-WG-8R	TCAAACGAAGTAGGCACCCAA		
PEDv-WG-9F	TGTTGGTGGTGCTGTCTGTAG	18529–20966	2438
PEDv-WG-9R	AGTCGCGCAGTAGCATTAGT		
PEDv-WG-10F	GAGGTGGTCATGGCTTTGAGA	20855–23380	2525
PEDv-WG-10R	CTGCCACAGAGCGACCATTA		
PEDv-WG-11F	ACACTGCAGCATGTAAGACCA	23075–25541	2466
PEDv-WG-11R	CGGTGACAAGTGAAGCACAG		
PEDv-WG-12F	GCTTTTCGTACTCTTTTTCCTGCT	25464–27820	2357
PEDv-WG-12R	ACCACTGGCTTACCGTTGTG		

* Based on the sequence of MEX/124/2014 (GB KJ645700).

**Table 2 viruses-16-00309-t002:** PEDV strains detected from 2013 to 2017 in Mexico and the type of strain in relation to the phylogeny of the PEDV genome.

Strain Name	Strain Type	Access Number
PEDV/MEX/MICH/01/2013	Pandemic 1	MH006957
PEDV/MEX/MICH/02/2013	Pandemic 1	MH006965
PEDV/MEX/MICH/19/2013	INDEL	MH006958
PEDV/MEX/GTO/01/2013	Pandemic 1	MH006959
PEDV/MEX/JAL/01/2014	Pandemic 1	MH006961
PEDV/MEX/JAL/02/2014	Pandemic 1	MH006965
PEDV/MEX/JAL/19/2014	Pandemic 1	MH004415
PEDV/MEX/VER/01/2014	INDEL	MH006960
PEDV/MEX/MICH/02/2015	INDEL	MH006962
PEDV/MEX/PUE/01/2015	Pandemic 1	MH004421
PEDV/MEX/VER/01/2015	Pandemic 1	MH013463
PEDV/MEX/VER/02/2015	Pandemic 1	MH011364
PEDV/MEX/SON/01/2015	Pandemic 2	MH013462
PEDV/MEX/PUE/01/2016	Pandemic 1	MH006963
PEDV/MEX/GTO/01/2016	Pandemic 1	MH004412
PEDV/MEX/JAL/03/2016	Pandemic 1	MH004413
PEDV/MEX/JAL/01/2017	Pandemic 1	MH004416
PEDV/MEX/JAL/02/2017	Pandemic 1	MH004417
PEDV/MEX/JAL/03/2017	Pandemic 1	MH004418
PEDV/MEX/JAL/04/2017	Pandemic 1	MH004419
PEDV/MEX/JAL/05/2017	Pandemic 1	MH004420
PEDV/MEX/JAL/19/2017	Pandemic 1	MH004419
PEDV/MEX/QRO/01/2017	Pandemic 1	MH013464
PEDV/MEX/QRO/02/2017	Pandemic 1	MH013466

## Data Availability

The sequences obtained in this study are available in GenBank under the following access numbers: MH006957, MH006965, MH006958, MH006959, MH006961, MH006965, MH004415, MH006960, MH006962, MH004421, MH013463, MH011364, MH013462, MH006963, MH004412, MH004413, MH004416, MH004417, MH004418, MH004419, MH004420, MH004419, MH013464, MH013466.

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
