# Peer review of "Phylogenetic and Molecular Analysis of the Porcine Epidemic Diarrhea Virus in Mexico during the First Reported Outbreaks (2013–2017)"

_viruses, 2024, doi:10.3390/v16020309_

Round 1
Reviewer 1 Report
Comments and Suggestions for Authors
The paper presents interesting phylogenic analysis of the PEDV strains detected in Mexico during the outbreaks 2013-2017. The authors presents additional analysis of the genome deletions and incidence on the structure of the spike protein with modeling supporting their findings.
The article is well written, supported by detailed review of the existing litterature about the topics. The material and method section is presenting clearly the techniques used to obtained the results. The result section is presenting the results of the phylogenetic analysis, investigation of mutations and modelisation of the spike protein of those mutants.
The findings are discussed in the discussion section with reference to epidemiological findings of Asian, European and American strains and clades.
I have no suggestion or comment for improvement.
Author Response
We thank the reviewer for his valuable comments.
Reviewer 2 Report
Comments and Suggestions for Authors
Molecular studies of PEDV in Mexico were already being conducted. The work does not bring much new. I suggest a to use for phylogenetic analysis the sequences of strains already detected in Mexico (other works also used samples from 2013-2016) and see if they have similar mutations.
Unfortunately, the images are unreadable and need to be corrected. What I miss in the discussion is a comparison of the PEDV sequences detected in this work to those described earlier in Mexico.
Feces and intestines were used as samples. The bioethics committee's approval for the collection of internal organs is lacking.
Specific comments:
1. please explain all abbreviations used in your work
2. line 135-135. 12.1% were INDEL strains. What about the rest of the samples?
3. Line 138. “The 12 PCR reactions were standardized” what does it mean? in materials and methods there is nothing mentioned about it
4. all trees are unreadable
5. lines 160-163 “ this analysis showed that the ……was probably an entry of European PEDV strains into Mexico” I don't understand this sentence
6. it is difficult to verify the described phylogenetic results because all the trees are unreadable
7. Figures 3 c and 3c should be bigger
8. Figures 3a and 3b are not very well readable
9. Figure 3a- the insertion LGL epitope SS” and SS6 are not visible
Round 2
Reviewer 2 Report
Comments and Suggestions for Authors
Unfortunately, the authors did not make corrections to the manuscript despite claiming so. The quality of the photos is poor and they are still unreadable. I also did not notice that earlier PEDV strains from Mexico were included in the phylogenetic analysis.
Author Response
Response to Reviewer 2 Comments.
Thank you very much for taking the time to review this manuscript. Please find the detailed responses below and the corresponding revisions/corrections highlighted/in track changes in the re-submitted files.
R1 round.
Comments Round 1.
Molecular studies of PEDV in Mexico were already being conducted. The work does not bring much new. I suggest a to use for phylogenetic analysis the sequences of strains already detected in Mexico (other works also used samples from 2013-2016) and see if they have similar mutations. Unfortunately, the images are unreadable and need to be corrected. What I miss in the discussion is a comparison of the PEDV sequences detected in this work to those described earlier in Mexico.
Response round 1.
We thank the reviewer for his suggestions to the manuscript. The information previously presented by Reveles-Félix et al., 2020 has been analyzed and has been included in the discussion of the results. Images and figures have been included separately so that they are properly visible.
Response round 2.
We have included the analysis of spike gene sequences reported in GenBank, belonging to Mexico. There are sequences in GenBank that our working group previously reported and correspond to 2013-2016 and are basically the same sequences that were integrated in the present report, where the complete PEDV genome has been included. However, for the analysis the sequences reported by García-Hernández et al (ref 23), Reveles-Félix et al (ref 24) and Vlasova et al (ref 24) are being included. This analysis is presented as a supplementary figure and its inclusion is mentioned in the text (lane 87-89; lane 132-135; lane 199-202).
Comments round 1.
Feces and intestines were used as samples. The bioethics committee's approval for the collection of internal organs is lacking.
Response round 1.
Information has been included in the M&M section. Veterinarians and pig farmers voluntarily provided the samples used in this study. The study was conducted according to the guidelines of and requirements indicated by the Institutional Bioethics Committee for the Care and Reasonable Use of Experimental Animals in Research Projects CENID-Animal Microbiology (Approval Code: CBCURAE-007-12/15/2015).
Response round 2.
All responses to comments have been included in the text box on the main page. Previously, an attached file had been sent, but it is better to do it this way.
Comments and responses round 1.
Specific comments:
- please explain all abbreviations used in your work
R: It has been corrected.
- line 135-135. 12.1% were INDEL strains. What about the rest of the samples?
R: Detailed information is presented.
- Line 138. “The 12 PCR reactions were standardized” what does it mean? in materials and methods there is nothing mentioned about it
R: This statement has been removed since it is not relevant in the presentation of results.
- all trees are unreadable
R: The figures are presented separately to make them more readable (Please review the attached files.).
- lines 160-163 “ this analysis showed that the ……was probably an entry of European PEDV strains into Mexico” I don't understand this sentence
R: The sentence has been corrected, seeking to make it more understandable.
- it is difficult to verify the described phylogenetic results because all the trees are unreadable. 7.Figures 3 c and 3c should be bigger. 8. Figures 3a and 3b are not very well readable. 9. Figure 3a- the insertion LGL epitope SS” and SS6 are not visible
R: The figures are presented separately to make them more readable.
We thank the reviewers and editors for their support.